# Assessing Upper Limb Function in Breast Cancer Survivors Using Wearable Sensors and Machine Learning in a Free-Living Environment

**DOI:** 10.3390/s23136100

**Published:** 2023-07-02

**Authors:** Nieke Vets, An De Groef, Kaat Verbeelen, Nele Devoogdt, Ann Smeets, Dieter Van Assche, Liesbet De Baets, Jill Emmerzaal

**Affiliations:** 1Department of Rehabilitation Sciences, KU Leuven, B-3000 Leuven, Belgium; nieke.vets@kuleuven.be (N.V.); an.degroef@kuleuven.be (A.D.G.); nele.devoogdt@kuleuven.be (N.D.); dieter.vanassche@kuleuven.be (D.V.A.); jill.emmerzaal@kuleuven.be (J.E.); 2CarEdOn Research Group, B-3000 Leuven, Belgium; kaat.verbeelen@uantwerpen.be; 3MOVANT Research Group, Department of Rehabilitation Sciences, University of Antwerp, B-2000 Antwerp, Belgium; 4Pain in Motion International Research Group, B-1000 Brussels, Belgium; 5Center for Lymphoedema, Department of Vascular Surgery, Department of Physical Medicine and Rehabilitation, UZ Leuven—University Hospitals Leuven, B-3000 Leuven, Belgium; 6KU Leuven, Department of Oncology, B-3000 Leuven, Belgium; ann.smeets@uzleuven.be; 7Surgical Oncology, UZ Leuven—University Hospitals Leuven, B-3000 Leuven, Belgium; 8Pain in Motion (PAIN) Research Group, Faculty of Physical Education and Physiotherapy, Department of Physiotherapy, Human Physiology and Anatomy, Vrije Universiteit Brussel, B-1000 Brussels, Belgium

**Keywords:** accelerometry, actigraphy, activities of daily living, breast neoplasms, functional activity, machine learning, upper extremity

## Abstract

(1) Background: Being able to objectively assess upper limb (UL) dysfunction in breast cancer survivors (BCS) is an emerging issue. This study aims to determine the accuracy of a pre-trained lab-based machine learning model (MLM) to distinguish functional from non-functional arm movements in a home situation in BCS. (2) Methods: Participants performed four daily life activities while wearing two wrist accelerometers and being video recorded. To define UL functioning, video data were annotated and accelerometer data were analyzed using a counts threshold method and an MLM. Prediction accuracy, recall, sensitivity, f1-score, ‘total minutes functional activity’ and ‘percentage functionally active’ were considered. (3) Results: Despite a good MLM accuracy (0.77–0.90), recall, and specificity, the f1-score was poor. An overestimation of the ‘total minutes functional activity’ and ‘percentage functionally active’ was found by the MLM. Between the video-annotated data and the functional activity determined by the MLM, the mean differences were 0.14% and 0.10% for the left and right side, respectively. For the video-annotated data versus the counts threshold method, the mean differences were 0.27% and 0.24%, respectively. (4) Conclusions: An MLM is a better alternative than the counts threshold method for distinguishing functional from non-functional arm movements. However, the abovementioned wrist accelerometer-based assessment methods overestimate UL functional activity.

## 1. Introduction

Upper limb (UL) dysfunctions are, with a prevalence of 60% after breast cancer surgery, common in breast cancer survivors [1,2,3,4,5]. From the viewpoint of breast cancer survivors, such upper limb dysfunctions are very impactful and lead to a high personal burden [6]. Indeed, upper limb dysfunction not only influences a woman’s independent performance of activities of daily living, it also negatively affects tasks at work, and leads to a decreased quality of life [6]. 

Adequately assessing the impact of treatment modalities on upper limb dysfunction and quality of life in breast cancer survivors is an emerging issue. Upper limb dysfunction is most frequently assessed using self-reported outcome measures, like the Disabilities of the Arm, Shoulder, and Hand Questionnaire (DASH) and its Short Form (Quick-DASH) [1,2,3,4,5,7], and the Shoulder Pain and Disability Index (SPADI) [4]. In breast cancer survivors, both the DASH and Quick-DASH have been reported to be valid in upper limb dysfunction evaluation [1,2,4,5,7]. Although self-reported outcomes are highly relevant, as they investigate the women’s perception of their upper limb dysfunction, there are several downsides. For example, they might be subject to response bias (i.e., a woman can score the items in such a way that she satisfies her treating clinician) or recall bias might occur (i.e., a woman may not remember what her actual movement behavior was like). Adding objective upper limb functional monitoring in daily life to self-reported outcomes could lead to a more comprehensive, all-encompassing upper limb assessment in breast cancer survivors. Wearable motion sensors could provide the ideal opportunity for this [8]. Upper limb wearables could clarify functional daily activities in different clinical populations, such as more recently obtained data in the post-stroke population, who often suffer from severe upper limb impairments [8,9,10,11]. 

To determine upper limb functioning from wearable sensors, the activity counts method is used as standard [11]. These activity counts quantify the intensity, duration, and when worn on both wrists, symmetry in arm use [12]. While this counts method is successful at standardizing research across populations and devices [13], recent research indicates that it overestimates the duration of upper limb use [9]. This overestimation appears to be mostly due to the inability of this activity counts method to classify functional (i.e., task-specific arm movements) from non-functional arm movement (e.g., arm swing while walking) [14]. Therefore, attempts have been made to enable the categorization of functional versus non-functional arm movement via machine learning models (MLM) [9,13]. Machine learning is a subgroup of artificial intelligence, based on mathematical algorithms, with the goal of making predictions based on identifying patterns in a data set. This MLM creates technological advances for example in cancer diagnosis [15]. The first results of integrating machine learning techniques in categorizing functional versus non-functional arm movements are promising [9,13]. A main downside, however, is that these models were trained, validated, and tested in either healthy persons [9] or persons after stroke [9,13]. They were also developed in standardized settings (e.g., occupational therapy centers), where variability in context is eliminated [9]. This means that it is currently unknown how these models can be generalized to real-world data.

It is of high clinical relevance to assess whether a learned model developed in healthy persons in a standardized setting can be generalized to a real-life setting (e.g., a home environment). Therefore, in this study, we will assess whether a model for the categorization of functional versus non-functional upper limb movements developed in healthy persons holds true in a clinical breast cancer population and whether this lab-based model also generalizes to a real-world environment. More specifically, in this study, we will assess the accuracy of a pre-trained lab-based MLM, developed by Lum et al. (2020) (which distinguishes functional from non-functional arm movements) [9] in breast cancer survivors’ home situations by analyzing the model’s output and video-recorded data. Lum et al. developed a lab-based decision tree machine learning model with an accuracy of 92.6% in detecting functional activity in healthy controls and individuals with stroke. The non-functional and functional activity prediction was made per 4-s epoch [9]. We hypothesize that the model’s accuracy will be lower in a home situation than in a lab-based environment. Additionally, we hypothesize that, in terms of upper limb use, the MLM will be closer to the ground truth data than the activity counts method.

## 2. Materials and Methods

### 2.1. Participants

Recruitment for this study was through a larger project at the University Hospitals Leuven (B-3000 Leuven, Belgium) about identifying the prognostic factors for the development of upper limb dysfunction in breast cancer survivors (UPLIFT) (clinicaltrails.com: NCT05297591). From the large UPLIFT cohort, a convenient sample of 10 breast cancer patients was recruited for the present study. The study was conducted by the Declaration of Helsinki, and the protocol was approved by the Ethics Committee Research UZ/KU Leuven (s66248). Ten participants, with or without upper limb impairments, were asked to participate. To be included, participants had to have had breast surgery at least one month prior. Participants were excluded if they had: distant metastases, a history of breast surgery, planned bilateral surgery, a neurological or rheumatological disease, and a cognitive and language functioning enabling incoherent Dutch communication between the examiner and the participant. Informed consent was obtained from all participants involved in the study.

### 2.2. Test Protocol in the Home Environment

The protocol used for the pre-trained lab-based MLM developed by Lum et al. (2020) was used in the present study, whereby participants were asked to perform four typical activities of daily living in their own home environment while wearing activity trackers and being video-recorded during the whole period [9]. For this study, we aimed to assess the pre-trained model in a home situation in breast cancer survivors. Therefore, the same activity instructions were given to the participants as described by Lum et al. [9].

Participants were instructed to perform the following four activities: laundry, kitchen, simulated shopping, and bed-making. All details about the performed activities are provided in Table 1. During the performance of these activities, participants wore two accelerometers (ActiGraph wGT3X-BT, sample frequency: 30 Hz, Actigraph Corporation, Pensacola, FL, USA), one on each wrist. Simultaneously the active participants were video recorded (Sony FDR-AX33, 25 fps). To mimic daily life as closely as possible, the women were instructed to perform the task as they would naturally. The video and accelerometer data were synchronized by a “calibration movement” consisting of three to five fast arm flexion movements in front of the camera. With the calibration movement, potential time delays between the sensors were detected and the signals were synchronized. Any data recorded before the first and after the second calibration was discarded. Furthermore, spot checks were performed to ensure synchronization was maintained throughout the data collection. In between activities, the participants were instructed to sit down while having a conversation with the researchers to increase the non-functional activity time. There is no limitation in the duration of performing the four activities.

### 2.3. Data Analysis

#### 2.3.1. Video Annotation, i.e., Ground Truth

The video data were analyzed for upper limb functioning by one researcher (JE). Upper limb functioning was defined by the Functional Arm Activity Behavioral Observation System (FAABOS) [10,16]. This observation system distinguishes functional (i.e., task-specific arm movements) from non-functional activities (e.g., arm swing while walking), and is considered the ground truth data for our comparison. When the annotator noted arm function, a timeline marker was placed in the video indicating whether the left, right, or both arm sides were functionally active. These timeline markers showed the functional activity’s beginning and end time points. All video data were annotated by frame into three categories. Category “0” meant “unknown” and included the calibration procedure. Category “1” were non-functional activities, including arm swings from walking and quiet sitting. Category “2” were functional activities and contained every movement that was considered functional by the FAABOS [10,16]. After removing the unknown data, the data were relabeled to either “0”, denoting non-functional arm activities, or “1”, denoting functional arm activities. The outcome ‘the total minutes of functional activity’ was calculated as the sum of all time spent in functional arm use per four seconds epoch.

Video data were annotated with Adobe Premiere Pro version 2023. After completing the video annotation, the timeline markers were exported to a CSV file for further analysis in MATLAB (version 2021b).

#### 2.3.2. Accelerometer Data Pre-Processing

From the accelerometer data, functional activity was defined using two separate methods: (1) the counts threshold method, first proposed by Uswatte et al. (2000) [11], and (2) a pre-trained MLM created by Lum et al. (2020) [9].

First, the acceleration data were pre-processed. The raw acceleration data in three directions were extracted from the native ActiGraph files using the Python module pygt3x (version 0.3.8). After these data were imported to MATLAB, the axes were redefined to match the configurations used by Lum et al. (2020) [9], i.e., they were defined in the anatomical position with the x-axis being the vertical axis (in line with the arm with the cranial direction being the positive direction); the y-axis being the medio-lateral axis (with a positive direction pointing medially for the right hand and laterally for the left hand); and the z-axis being the anterior-posterior axis (with a positive direction pointing anteriorly) (Figure 1 and Figure 2).

#### 2.3.3. Counts Threshold Method

Activity counts were calculated from the raw acceleration data using the Python package agcounts (version 0.1.7), developed by Neishabouri et al. (2022) [17]. The sampling frequency was kept at 30 Hz, and the to the epoch size was 1 s. We chose this method over the ActiGraph software to ensure repeatability without needing proprietary software. The outcome ‘total minutes active’ was calculated as the sum of the 1 s epochs where the counts threshold exceeded 1, according to the previously used method by Lum et al. [9].

#### 2.3.4. Machine Learning Pipeline

To match the sampling frequency used in the pre-trained MLM, we resampled our acceleration data from 30 Hz to 50 Hz using spline interpolation. After resampling the data, further processing was carried out following the processing protocol of Lum et al. (2020) [9]. Each 4 s epoch of the acceleration data was assigned a label based on video annotation. The labels were functional, non-functional, mixed, or unknown, with the majority of frames (i.e., 50%+) in the epoch determining the label. Any data that were labeled unknown were discarded from further analysis.

From the accelerometer data, 11 features were calculated for each 4 s epoch [9]. These included the mean and variance of the acceleration in the x, y, and z directions, as well as the mean, variance, minima, maxima, and Shannon entropy of the Euclidean norm across the epoch. All features were normalized using min-max scaling. The calculated features were used in a pre-trained model to predict functional and non-functional blocks. The classification accuracy was defined as the percentage of data correctly classified into the functional and non-functional categories, using the video-annotated data as ground truth. Considering we have an unbalanced data set, we also considered recall, specificity, and f1-score. These outcomes were analyzed using the methodology described below, and were based on primary terminologies, illustrated in Table 2 [18,19].

The recall is defined as the percentage of functional activity that we predicted correctly out of all sections of actual functional activity. In this case, recall indicates what percentage, out of all functional activity, is, in fact, functional activity.
$$\mathrm{recall}=\frac{TP}{TP+FN}$$

Specificity is the percentage of correctly labeled sections of non-functional activity out of all sections of actual non-functional activity. Therefore, specificity indicates what percentage, out of all non-functional activity, is, in fact, non-functional activity.
$$\mathrm{specificity}=\frac{TN}{TN+FP}$$

The f1-score is the harmonic mean of precision and recall. Precision is defined as the percentage of correctly predicted functional activity out of all sections labeled functional activity (the true and false positives), where the formula for precision = (*TP*/*TP* + *FP*). This harmonic mean is best when there is a balance between the two. The f1-score is a value between 0 and 1, with 0 being the worst score and 1 the best score. If the score is lower than 0.5, the harmonic mean is not good.
$$f1\mathrm{-score}=2\times \frac{\mathrm{recall}\times \mathrm{precision}}{\mathrm{recall}+\mathrm{precision}}=2\times \frac{\frac{TP}{TP+FN}\times \frac{TP}{TP+FP}}{\frac{TP}{TP+FN}+\frac{TP}{TP+FP}}$$

Finally, the number of active minutes was calculated using the predicted functional blocks. This was done by multiplying the number of functional blocks by 4 s and dividing by 60.

### 2.4. Outcomes

Seven outcome parameters were considered in this study: (1) the prediction accuracy, recall, specificity, and f1-score of the pre-trained MLM; (2) the total minutes of functional activity based on the video-annotated data, i.e., ground truth; (3) the total minutes of functional activity predicted by the pre-trained MLM; (4) the total minutes of functional activity as calculated from the counts threshold method; (5) the percentage in functional activity following video-annotated data, i.e., the ground truth; (6) the percentage in functional activity predicted by the pre-trained MLM; (7) the percentage in functional activity calculated with the counts threshold method, whereby the percentage in functional activity represents the functional activity relative to the total activity time.

### 2.5. Statistical Analyses

Furthermore, the relationship between prediction accuracy of the pre-trained MLM and the QuickDASH score (range 0–100) was investigated via a Spearman correlation test (with a two-sided *p*-value of <0.05 considered statistically significant). To investigate if upper limb dysfunctions are presented, does this result in lower accuracy levels?

## 3. Results

### 3.1. Demographic Data

An overview of the characteristics of the ten women participating in this study is provided in Table 3. Originally, eleven participants signed the informed consent form; however, only data from ten participants could be used for analysis. One participant had to be excluded due to corrupted accelerometry data. The age of the participants ranged from 42 to 71 (median 50.5 IQR [43.8–56.0]), and the median BMI was 25.7 (IQR [22.6–28.6]). All ten participants were treated with surgery, six underwent a mastectomy, four underwent breast-conserving surgery, seven had a sentinel lymph node biopsy, and three had an axillary lymph node dissection. Six women were operated on their dominant side, three received neo-adjuvant chemotherapy, one finished adjuvant chemotherapy, and six finished radiotherapy. The median QuickDASH score is 11.4 (IQR [3.38–15.9]), whereby a score of more than 15 means that there are upper limb dysfunctions presented (which was the case for three women). The average duration of the activity monitoring was 25 min. A plot representing typical video annotated data and raw accelerometry data is added to the Appendix A. 

### 3.2. Prediction Accuracy

Table 4 shows the prediction accuracy, recall, specificity, and f1-score for upper limb functioning in ten breast cancer survivors for the left and right arm following the pre-trained MLM [8]. Good accuracy is presented for the pre-trained MLM [9]. The recall, which indicates that we were correct for all sections in which we predicted actual functional activity, seems to indicate a good result for the pre-trained MLM [9]. The same good result is presented for the specificity, indicating that we were correct for all sections where we predicted actual non-functional activity. However, a poor f1-score is presented. Therefore, the good recall and specificity score come at the expense of the number of false positives.

### 3.3. Minutes Active and Percentage Functional

In Table 5, the ‘total minutes of functional activity’ and ‘percentage in functional activity’ from the video-annotated data (ground truth), pre-trained MLM, and the counts method are displayed for the left and right arm. These results show that there is still an overestimation of the functional activity of the upper limb in breast cancer survivors. The differences between the video-annotated data and the functional activity following the pre-trained MLM range between 0.08 and 0.23 percent, and the mean is 0.14 (±0.04) for the left and 0.10 (±0.04) for the right side. For the video-annotated data versus the counts threshold method, differences are even larger (ranging between 0.14 and 0.38 percent) with a mean of 0.27 (±0.07) for the left and 0.24 (±0.07) for the right side. Figure 3 displays a scatter plot of the percentage functional activity in comparison to the ground truth data.

### 3.4. Correlation Accuracy and DASH Scores

A Spearman correlation test was performed between the QuickDASH scores and the prediction accuracy for functional activity of the left and right arm side (rho = 0.544 and = 0.477). The two-sided *p*-value was respectively *p* = 0.104 and *p* = 0.163, indicating no relationship between the accuracy and upper limb dysfunctions.

## 4. Discussion

To the best of our knowledge, this is the first study examining the accuracy of an MLM for defining functional upper limb activity in breast cancer survivors in a home situation by comparing the MLM with video-annotated data (i.e., ground truth). The MLM used was based on the MLM developed by Lum and his colleagues in healthy persons and persons after stroke [9]. Additionally, a comparison with an activity counts method was made. The mean difference in percentage of functional activity is, between the video-annotated data and the functional activity following the pre-trained MLM, was 0.14 (±0.04) for the left and 0.10 (±0.04) for the right side. For the video-annotated data versus the counts threshold method, the mean differences are 0.27 (±0.07) percentage for the left and 0.24 (±0.07) percentage for the right side. The results show that this MLM evaluates upper limb functioning more accurately than the commonly used counts method. However, the MLM gives an overestimation of upper limb functioning when compared to the video-annotated data (i.e., ground truth). An overestimation is present in both methods for the prediction of functional activity compared to the video-annotated data. However, the overestimation in the prediction of functional activity based on the counts method was far greater than the estimation based on the MLM. This gives rise to the question of the validity of the counts threshold method for assessing upper limb functioning.

Lum et al. developed the MLM algorithm in a laboratory setting and found a correlation with the ground truth of r = 0.99, with an accuracy of 92.6% in the paretic limb of stroke patients [9]. Our results indicate poorer accuracy scores (i.e., range from 0.77 to 0.90), despite using the same MLM. This could be due to several reasons. First, Lum et al. developed and fine-tuned the MLM in data collected from ten healthy controls and ten individuals with stroke [9]. Based on Fisher’s findings, we assumed that breast cancer survivors’ amount of functional upper limb use would not differ significantly from healthy controls, and therefore no loss in accuracy of the MLM developed by Lum et al. will present in our population [8,9]. Unfortunately, our results do not comply with the accuracy results described by Lum et al. [9].

In this study, no relation was found between self-reported upper limb dysfunction in breast cancer survivors and the MLM’s accuracy. However, upper limb impairments may contribute to the accuracy results of the MLM, for example, the amount of use, could be different if there are UL dysfunction presented. On the other hand, a limitation of this study is the rather small sample size. A larger sample size and more variability in the presence of UL dysfunctions might be essential to determine the consistency of MLM in healthy participants and women with upper limb dysfunction following breast surgery, to further determine the influence of the condition on the accuracy. Second, even though the accuracy, recall, and specificity were relatively high, average of >75%, the f1-score was low. This indicates that the model prioritizes the prediction of functional activity over the precision of the model, leading to a large number of false positives. Third, and of utmost importance, in this study, the MLM’s accuracy was tested outside a controlled lab setting. While performing this kind of assessment in an ecologically more valid environment, it is to be expected that unforeseen activities with the arm happen, e.g., opening/closing doors, or switching lights on/off. Another clear example of additional functional movements that were observed in the home situation was the arm movements needed for taking the stairs, or going to the car or the sleeping room to perform the required study task. These kinds of activities were not included in the validation protocol of Lum [9]. However, holding the stair railing when taking the stairs was considered a functional activity and thus rated as such when annotating the video data. Additionally, since we tested the women in their own home environment, the context was different for each individual. This led to a decrease in task standardization. For example, in the lab setting, each object was placed at the same location for each participant (e.g., on a shelf in a fridge). In contrast, in the home environment, some women had their fruit in a fridge while others stored them in a cellar. Therefore, although the task description was similar to the task description used by Lum and colleagues (2020), the context was different [9]. We suggest that the context, especially, plays an important role in the accuracy, since the results of the women’s QuickDASH scoring were relatively low (meaning that upper limb dysfunctions were limited), plus the fact that there was no correlation between the quickDASH scoring and the prediction accuracy.

To evaluate recovery after breast cancer treatment, the real-world impact and assessment of daily life movement behavior is important [14]. Functional task performances in a lab setting can be influenced by breast cancer surgery [20,21]. Besides, we know from previous research that the level of activity can have a positive effect after breast cancer treatment and that an active approach is effective for treating post-operative pain and impairments in the shoulder range of motion [22,23]. However, as clinical-based assessments can indicate what a person’s capabilities are, this does not reflect what a patient actually does in daily life [14]. Therefore, the importance of assessing functional upper limb movement in breast cancer survivors in their home situation needs to be highlighted [12,20].

Although the MLM shows promising results for future research, it is too early for a direct transfer to an unconstrained environment, without a reduction in accuracy. Therefore, correctly and objectively quantifying functional upper limb activity outside of a lab setting remains a research priority. The MLM developed by Lum et al. (2020) [9] used a random forest classifier model to determine the label of functional or non-functional upper limb activity. Adaptations to this MLM so that it can be transferred to an unconstrained environment are thus rather difficult. Therefore, as a next step, we suggest improving the MLM’s accuracy in the home environment and exploring the use of newer machine learning technics, such as deep learning neural networks.

## 5. Conclusions

The results of this study show that the machine learning model developed by Lum et al. more accurately evaluates upper limb functioning than the commonly used counts method. Interpreting the results of upper limb functioning with the use of the standard counts threshold method of accelerometry data should therefore be undertaken with caution. The error in terms of accuracy is large, since a large amount of non-functional activity is interpreted as functional activity. A machine learning model is a better alternative, as its results are closer to the video data, but an overestimation is still present. Future research should be done in developing a more accurate machine learning algorithm to objectify upper limb functioning in clinical populations such as breast cancer survivors in daily life outside of a controlled setting.

## Figures and Tables

**Figure 1 sensors-23-06100-f001:**
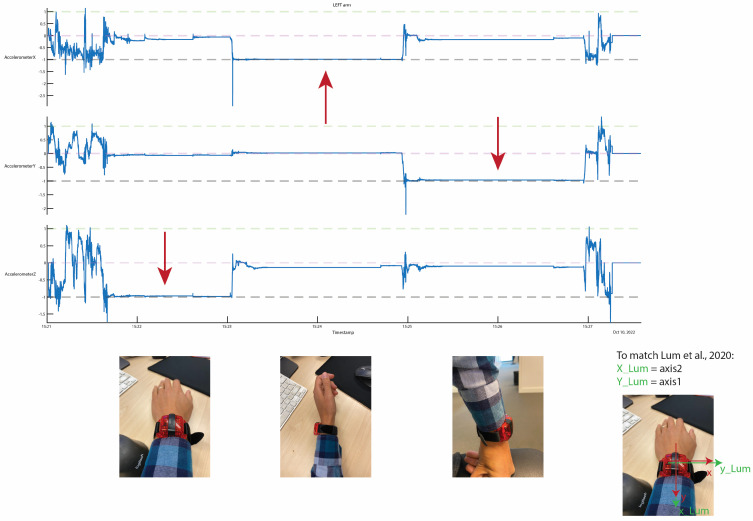
Redefining of the axis to match the configurations used by Lum et al. for the left side [9]. The red arrows indicate the gravitational acceleration.

**Figure 2 sensors-23-06100-f002:**
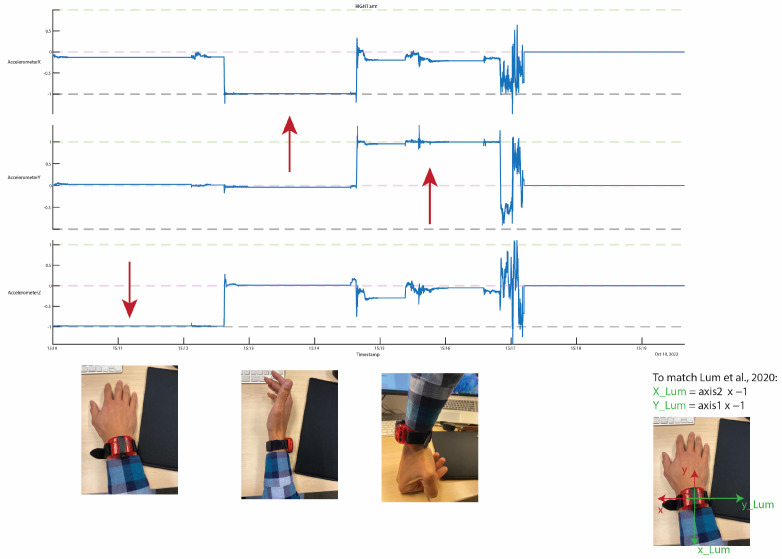
Redefining of the axis to match the configurations used by Lum et al. for the right side [9]. The red arrows indicate the gravitational acceleration.

**Figure 3 sensors-23-06100-f003:**
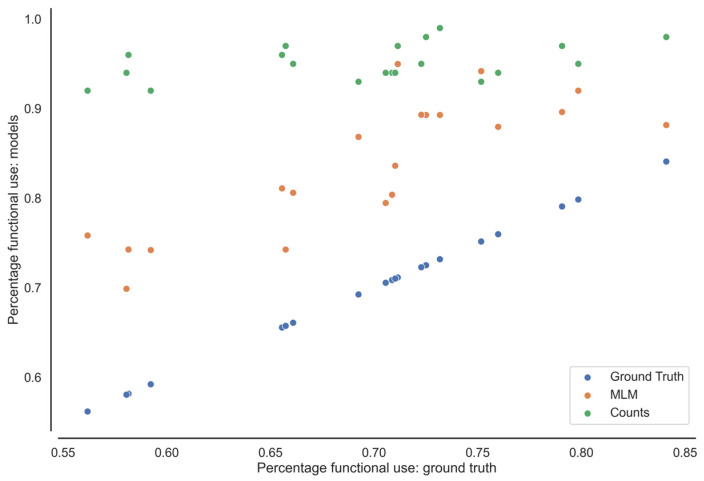
The percentage of functional activity from the MLM and counts threshold method (Counts) compared with video-annotated data (Ground Truth) from the left- and right-side arm of each participant.

**Table 1 sensors-23-06100-t001:** Description of the performed activities.

Activities	Description
Laundry activity	Participants were instructed to (1) move clothes from a closet or basket into a washer and close the washer, (2) remove the clothes from the washer, put them in the dryer, and close the door, and (3) remove the clothes from the dryer and fold them or hang them back in the closet.
Kitchen activity	Participants were instructed to (1) load and unload four or five items from the dishwasher, (2) cut an apple, (3) pick up one item from the floor, and (4) use a broom or a dust mop for home to sweep the floor.
Shopping activity	Participants were instructed to (1) gather four or five items out of the supply closet in their grocery store bag or box, (2) place them into the car, step into the car, then step out, and remove the groceries from the car, and (3) put the groceries back in the supply closet.
Bed making activity	Participants were instructed to (1) remove the sheets and pillowcases from their bed and (2) replace them.

**Table 2 sensors-23-06100-t002:** The primary terminologies used for the analyses of the recall, specificity, and f1-score, visualized.

		Reality	
		Functional Activity	Non-Functional Activity
Prediction	Functional activity	True Positive (TP)	False Positive (FP)
	Non-functional activity	False Negative (FN)	True Negative (TN)

**Table 3 sensors-23-06100-t003:** Demographic data of the participants.

Subj ID	Age (Years/Old)	BMI (kg/cm^2^)	Operated Side	Surgery	(Neo-)Adjuvant Treatment	QuickDASH Score
P_001	44	31.87	R	ME + SN	TAM.	0
P_002	48	19.69	L	ME + SN	Adj. CT +TAM	9.1
P_003	50	25.09	R	BCS + SN	Adj. RT + TAM	38.6
P_004	53	29.29	L	ME + SN	/	4.5
P_005	52	26.29	L	BCS + SN	Adj. RT + TAM	13.6
P_006	45	24.6	L	ME + ALND	Neo-adj. CT +Adj. RT +AI	11.4
P_007	52	27.88	R	ME + ALND	Neo-adj. CT +Adj. RT	15.9
P_008	43	23.52	R	ME + SN	TAM	0
P_009	65	28.37	R	BCS + SN	Adj. RT + AI	15.9
P_010	72	19.83	L	BCS + ALND	Neo-adj. CT +Adj. RT + AI	11.4
Median [IQR]	50.5 [43.8–56.0]	25.7 [22.6–28.6]				11.4 [3.38–15.9]

Abbreviations: P, participant; L, left; R, right; ME, mastectomy; BCS, breast-conserving surgery; SN, sentinel lymph node biopsy; ALND, axillary lymph node dissection; Neo-adj., neo-adjuvant treatment; Adj., adjuvant treatment; CT, chemotherapy; RT, radiotherapy; TAM, tamoxifen; AI, Aromatase-inhibitor; IQR, interquartile range.

**Table 4 sensors-23-06100-t004:** Prediction accuracy, recall, specificity, and f1-score of the ten breast cancer survivors.

	Left Arm				Right Arm			
Subj ID	acc	Recall	Spec	f1	acc	Recall	spec	f1
P_001	0.82	0.96	0.80	0.47	0.82	0.75	0.83	0.45
P_002	0.82	1.00	0.80	0.53	0.87	0.94	0.86	0.63
P_003	0.90	0.89	0.90	0.30	0.89	0.30	0.93	0.25
P_004	0.83	0.94	0.83	0.32	0.83	0.87	0.82	0.57
P_005	0.77	1.00	0.76	0.23	0.78	0.74	0.79	0.45
P_006	0.80	0.85	0.80	0.31	0.88	0.82	0.89	0.55
P_007	0.85	1.00	0.84	0.36	0.88	1.00	0.87	0.63
P_008	0.81	1.00	0.80	0.33	0.83	0.72	0.84	0.44
P_009	0.88	0.95	0.88	0.55	0.86	0.75	0.86	0.47
P_010	0.83	1.00	0.82	0.12	0.81	0.75	0.82	0.32
avg	0.83	0.96	0.82	0.35	0.85	0.77	0.85	0.47

Abbreviations: P, participant; spec, specificity; f1, f1-score; avg, average.

**Table 5 sensors-23-06100-t005:** The total minutes of functional activity and the percentage functionally active for each participant following video-annotated data (ground truth), the machine learning model (MLM), and the counts threshold method (counts threshold). The mean difference indicates the mean of the difference with the ground truth (mean [SD]).

	Left Arm			Right Arm		
Subj ID	Ground Truth	MLM	Counts Threshold	Ground Truth	MLM	Counts Threshold
Total minutes of functional activity						
P_001	15.93	19.73	26.28	16.80	19.60	26.22
P_002	12.33	15.47	16.60	13.53	15.67	16.70
P_003	21.87	24.27	26.88	23.27	23.93	27.13
P_004	19.73	23.80	28.43	17.33	20.60	27.98
P_005	12.53	16.53	17.05	12.73	15.47	16.82
P_006	12.20	15.13	20.27	13.80	15.27	20.50
P_007	10.40	12.33	15.15	11.27	12.93	14.87
P_008	16.07	20.07	19.77	17.07	19.60	20.35
P_009	15.40	17.47	20.33	15.33	17.27	20.32
P_010	14.40	17.47	19.42	14.53	17.47	19.57
Percentage of functionally active						
P_001	0.56	0.69	0.92	0.59	0.69	0.92
P_002	0.69	0.87	0.93	0.76	0.88	0.94
P_003	0.79	0.87	0.97	0.84	0.86	0.98
P_004	0.66	0.80	0.95	0.58	0.69	0.94
P_005	0.71	0.94	0.97	0.72	0.88	0.95
P_006	0.58	0.72	0.96	0.65	0.72	0.97
P_007	0.66	0.78	0.96	0.71	0.82	0.94
P_008	0.75	0.94	0.93	0.80	0.92	0.95
P_009	0.71	0.80	0.94	0.71	0.79	0.94
P_010	0.73	0.88	0.98	0.73	0.88	0.99
Mean difference [SD]		0.14 [0.04]	0.27 [0.07]		0.10 [0.04]	0.24 [0.07]

## Data Availability

Data is unavailable due to privacy and ethical restrictions.

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
