# Peer review of "Assessing Upper Limb Function in Breast Cancer Survivors Using Wearable Sensors and Machine Learning in a Free-Living Environment"

_sensors, 2023, doi:10.3390/s23136100_

Round 1

Reviewer 1 Report

 The aim of this study is to determine the accuracy of a pre-trained lab-based machine learning model (MLM) to distinguish functional from non-functional arm movements in a home situation in breast cancer survivors (BCS). The data was collected for participants who performed four daily life activities while wearing two wrist accelerometers and being video recorded. To define upper limb UL functioning, video data was annotated and accelerometer data were analyzed using a counts threshold method along with a MLM. In addition, prediction accuracy, recall, sensitivity, f1scor, total minutes of functional activity, and percentage of functional activity were included in the procedure.

The results suggest good MLM accuracy, recall, and specificity, while the f1-score is reduced. An overestimation of the total minutes of functional activity and ‘percentage of functional activity was found via the MLM. Between the video-annotated data and the functional activity determined by the MLM, mean differences were 0.14% and 0.10%

For the video-annotated data versus the counts' threshold method, mean differences are 0.27% and 0.24%, respectively

The findings led to the conclusion that the MLM is a better alternative than the counts' threshold method to distinguish functional from non-functional arm movements. Nevertheless, an overestimation of the UL functional activity occurred.

This work uses state–of–the–art methodology and data analysis to determine the accuracy of a pre-trained lab-based machine learning model (MLM) concerning breast cancer survivor research. The methodology is known, clear, and correctly applied.

The paper is well written and the authors provided a good presentation of the issue in question, while the findings add to relevant literature.

In my opinion, the paper can be published in the present form

a minor editing might be usuful

Author Response

Thank you for these kind words!

The paper was reviewed for spelling, punctuation, and grammatical errors to improve the quality of the English language.

Reviewer 2 Report

Review

The manuscript, “Assessing upper limb function in breast cancer survivors using wearable sensors and machine learning in a free-living environment” by Vets et al. assessed whether a model for the categorization of functional versus non-functional upper limb movements, developed in healthy persons, holds true in a clinical breast cancer population and whether this lab-based model also generalizes to a real-world environment.

Comments and Suggestions for Authors:

1- Please correct the keywords according to MeSH.

2- Every succeeding paragraph should be indented.

3- In lines 85 to 87, explain more about Lum's study.

4-It is better to add the general results of "Demographic data" to Table 2.

5-In line 98, it is said that 10 people entered the study, while the results show information about 11 people. Please explain about this.

Author Response

We want to thank the reviewer for these comments. We addressed the remarks as best as possible in the attachment. Further, the paper was reviewed for spelling, punctuation, and grammatical errors to improve the quality of the English language.

Please see the attachment for the responses. 

Reviewer 3 Report

Related “Article

Assessing upper limb function in breast cancer survivors using wearable sensors and machine learning in a free-living environment.

Comments

1-Upper limb dysfunction is 50most frequently assessed using self-reported outcome measures, like the Disabilities of 51 the Arm, Shoulder, and Hand Questionnaire (DASH) and its Short Form (Quick-DASH) and the Shoulder Pain and Disability Index (SPADI). In breast cancer survivors, both the DASH and Quick-DASH are reported to be valid in upper limb dysfunction evaluation.

Authors should clearly explain these methods and why Shoulder Pain and Disability Index (SPADI) is not valid in this case. In other words, the authors explain the advantages and limitations of these methods. It would be more appropriate to present a table.

2- References 1, 5 and 9 should be replaced with more recent studies.

3. The word “Machine learning” should be added in keywords.

4. Machine learning is a branch of artificial intelligence (AI) and computer science that focuses on the use of data and algorithms to imitate the way that humans learn, gradually improving its accuracy. The authors should insert a separate section on Machine learning in medical approaches in the main text.

5. “An overview of the characteristics of the ten women participating in this study is provided in table 2”. The large number of samples is considered a bonus for research studies. Is it possible to add more samples?

6. In the "Discussion" section, the authors have referred to only one reference. It is suggested to discuss at least 5 more studies. Certainly, studies consistent with the results obtained in this manuscript will be more interesting.

7. The conclusion section is incomplete and needs to be expanded.

Author Response

(The authors gave the same response as above.)

Reviewer 4 Report

In this manuscript, the authors proposed a machine learning model (MLM) for interpreting results of upper limb functioning. They found that the MLM is a better alternative than the counts threshold method to distinguish functional from non-functional arm movements. Although the as-obtained results are closer to the video data, an overestimation is still present. Specific comments,

(1)    Two accelerometers were used in this study. The difference between two accelerometers should be addressed.

(2)    How long have the patients been monitored?

(3)    Typical video data and curves recorded by accelerometers should be provided, for instance, as supplementary materials.

(4)    Minor point, the sentences ‘Indeed, the MLM detects a large percentage of move- 307 ment as functional instead of non-functional in the free-living environment. However, the overestimation of upper limb functioning following the MLM is smaller than by the use of the conventional counts threshold method.’ at the end of conclusion are difficult to understand.

Moderate editing of English language required.

Author Response

(The authors gave the same response as above.)

Round 2

Reviewer 2 Report

Thank you very much for corrections.

Reviewer 3 Report

Accepted in present form.